# Lamb and Wool Provisioning Ecosystem Services in Southern Patagonia

**Pablo Luis Peri** [1,2,*] , **Yamina M. Rosas** [3] , **Emilio Rivera** [1] **and Guillermo Martínez Pastur** [3]

1 Instituto Nacional de Tecnología Agropecuaria (INTA), Río Gallegos 9400, Argentina; rivera.emilio@inta.gob.ar
2 CONICET—Universidad Nacional de la Patagonia Austral (UNPA), Río Gallegos 9400, Argentina
3 Laboratorio de Recursos Agroforestales, Centro Austral de Investigaciones Científicas (CADIC CONICET), Ushuaia 9410, Argentina; yamicarosas@gmail.com (Y.M.R.); cadicforestal@gmail.com (G.M.P.)
* Correspondence: peri.pablo@inta.gob.ar

**Abstract:** In Southern Patagonia, grasslands are the principal food resource for sheep reared for meat and wool as the main provisioning ecosystem services (ES). The main objective of this study was to model lamb and wool production as provisioning ES at a regional scale using climatic, topographic, and vegetation variables from sheep farms across Santa Cruz province. At a regional level, animal yield ranged from 0.25 to 0.69 g lamb/m$^2$/yr and 0.10 to 0.19 g greasy wool/m$^2$/yr. We used multiple regression models to produce maps of lamb and wool provisioning ES across Santa Cruz province. The model for variation of lamb production explained 96% of the variance in the data and the most significant predictor variables were temperature seasonality, normalized vegetation index (NVDI, dimensionless), and desertification index. The most important variables for the model of greasy wool production were isothermality, temperature seasonality, and NVDI, which together explained 98% of the variance. The lowest CF values of both products (lamb and wool) were located in more productive grasslands. There were differences in lamb and wool production across vegetation types with the highest values being located in more productive grasslands (0.51 g lamb/m$^2$/yr in *Nothofagus antarctica* forest and 0.15 g greasy wool/m$^2$/yr in Magellanic grass steppe and *N. antarctica*). Lamb and greasy wool yields decreased with desertification gradient due to erosion processes. The main limitation of the model is related to the data availability at landscape level, which must be improved in future studies by accounting for soil type, fertility, and soil water content. The results of lamb and wool production found in the present work assist in characterizing the provisioning ES ecosystem of livestock products in Southern Patagonia. The successful management of livestock becomes an important challenge to the commercial and policy communities to satisfy society's need for food and wool products under sustainable grassland management.

**Keywords:** rangeland; livestock; climate; lamb production; wool production

## 1. Introduction

Grasslands in Southern Patagonia are a major land cover type that have implications for the provision of diverse ecosystem services (ES) and human wellbeing. In Southern Patagonia (Santa Cruz province), extensive livestock production reared for meat and wool is the main agricultural activity based on natural grasslands where continuous grazing with fixed stocking rates in large paddocks (1000 to 5000 ha) prevails over grazing systems subjected to regular evaluations and rotational rests [1,2]. The productivity of Patagonian sheep herds is strongly dependent on environmental and management factors that affect reproductive efficiency and animal performance in the areas of genetics, animal health, time period and type of mating, assistance during lambing, shearing practices, and nutrition synchronization between forage supply and demand [3–6].

Livestock play an important role in the provision of ES by transforming grass and herbs into nutritious foods and useful products for human consumption (milk, meat,

wool). Livestock provide one-third of humanity's protein intake and 13 percent of all calories [7]. However, there is a lack of information in Patagonia about spatially explicit livestock provisioning ES assessments that is used to support decision-making [8]. Previous studies at a regional level have reported results on provisioning ES of timber from native forests [9], livestock and firewood from silvopastoral systems [10], regulating ES such as soil carbon [11] and nitrogen content [12], and some studies have analyzed cultural ES at a landscape level [13,14]. Here, we evaluated the importance of livestock in providing societies with food and wool as provisioning ecosystem services that determine incomes and employment in areas like Patagonia.

Long term intensive grazing has markedly reduced vascular plant diversity and cover, decreased the availability and desirability of forage, facilitated the encroachment of invasive and exotic species, and increased soil degradation and desertification within Patagonian rangelands [15–17]. Thus, heavy and unsustainable grazing conditions threaten the future of livestock productivity, therefore threatening the long-term health and wellbeing of the local economy. In this context, rangeland management should be based on maintaining the capacity of socio-ecological systems to provide food and services for current and future human needs by maintaining biodiversity and regulating and supporting ES (e.g., carbon and nitrogen stocks in the ecosystems). In fact, negative consequences (e.g., desertification) [18–20] due to overgrazing [17,21], land use conversion and climate changes [19] have been reported for the steppe ecosystem. According to climate models, mean maximum annual temperature is predicted to increase by 2 to 3 °C by 2080 in the latitudinal range of 46° to 52° S [22], therefore presenting additional challenges for the future. The projected changes within climatic variables will most likely have profound effects on ES, biodiversity, and land use capacity throughout the globe, including Patagonia.

The main objective of the present work was to model lamb and wool production as provisioning ES at a regional scale using climatic, topographic, and vegetation variables from sheep farms across Santa Cruz province. We hypothesized that lamb production would be more sensitive than wool production to harsh environmental conditions (low soil moisture conditions) and forage quantity at the regional scale in Patagonia.

## 2. Material and Methods

### 2.1. Characterization of the Study Area

For this study, from the PEBANPA (Parcelas de Ecología y Biodiversidad de Ambientes Naturales en Patagonia Austral—Biodiversity and Ecological Long Term Plots in Southern Patagonia) network [21] we selected 120 permanent plots across Santa Cruz province (Figure 1A) to estimate the lamb (gr lamb/m$^2$/yr) and greasy wool (gr greasy wool/m$^2$/yr) yields as grasslands provisioning ecosystem services. These plots are located in five ecosystem categories (Mata Negra shrubland, Dry Magellanic steppe, Humid Magellanic steppe, Central Plateau grasslands, and Andean grasslands) and desertification conditions (Figure 1B,C). Further details about environmental conditions across Santa Cruz Province can be found in Peri et al. [21].

In the region, annual rainfall ranges from 800–1000 mm/year in the Andes Mountains (west) and decreases to 200 mm/year in the eastern part of Santa Cruz Province. The mean annual precipitation to potential evapotranspiration ratio of the steppes fluctuates between 0.45 and 0.11, with marked soil water deficits in summer. The variations in local topographic and edaphic characteristics, combined with a significant precipitation gradient, substantially influence the grasslands' forage production. Mean annual temperatures range between 5.5 and 8.0 °C. The windiest season occurs between November and March, producing frequent and severe south-southwesterly wind storms reaching over 80 km/h.

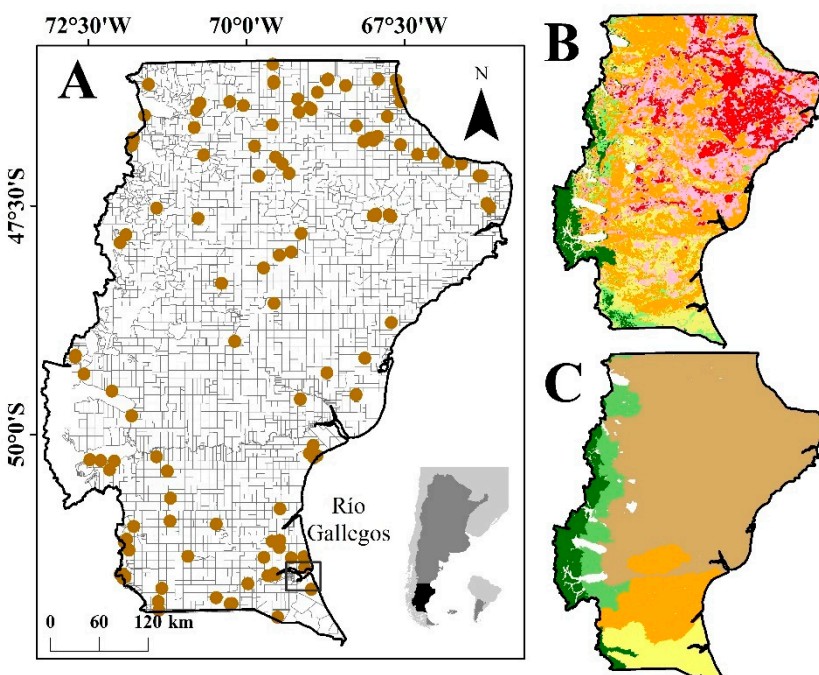

**Figure 1.** Study area. (**A**) Location (light grey = South America, dark grey = Argentina, black = Santa Cruz), sample sites (brown dots), and provincial farms (680 productive sheep farms); (**B**) desertification index (dark green = none, light green = slight degraded, yellow = moderate desertification, orange = moderate to severe desertification, pink = severe desertification, red = very severe desertification [18]; (**C**) ecological areas (brown = central plateau, orange = mata negra thicket, yellow = Magellanic grass steppe, green = *Nothofagus* forest, light green = Sub-Andean vegetation).

The main activity in the evaluated sites is extensive sheep production, mostly with the Corriedale breed. The animals use different paddocks from May to September (mating and gestation), September to January (lambing and lactation), and January to May (from weaning to mating). These paddock changes are associated with specific activities, such as eye-shearing (May), pre-lambing shearing (September), and marking (January). Sheep are eye-shorn to facilitate the visibility of the animals to feed. Paddocks situated above 700 m above sea level are mostly used in summer, because they are covered with snow during the winter season. Lamb production implies a particular nutritional requirement curve, with higher demand before the start of winter to ensure pregnancy (May mating) and during winter until spring regrowth. The months before spring regrowth are critical because they coincide with the last two-month period of sheep gestation, when nutritional requirements increase considerably [23]. Ewes must recover after lambing until May, but this coincides with the lactation period, which is generally interrupted due to weaning in January or February.

### 2.2. Characterization of Sheep Production

The farm areas in this study range from 20,000 to 35,000 ha with a breeding ewe flock size of 5000–22,500 head/farm. The estimation of carrying capacity is based on the biomass production of short grasses and forbs that grow in the space among tussocks of each ecosystem and the requirements of 530 kg DM/yr for 1 Corriedale ewe of 49 kg live weight, which represents a "Patagonian sheep unit equivalent (PSUE)" [23]. The stocking rate varied from 0.20 to 0.75 PSUE/ha. The average live weight of lambs sold fluctuates between 20 and 25 kg and greasy wool is 4.2–5.0 kg/animal. Lambing as a productivity indicator fluctuates between 70 and 90%, and the lamb growth rate from birth to finishing after 100 days is between 170 and 200 g/day [24].

At each sampling location during the period 2016–2018, plant forage production and quality were measured in a 20 m × 50 m quadrat (1000 m$^2$). Above ground net primary productivity (ANPP) of plant forage (grasses and graminoids) was obtained from destructive sampling at peak biomass, which occurs from December–January. This was done by clipping peak live plant material (current year's green production, excluding woody tissue) obtained from three randomized 0.2 m$^2$ in three permanent enclosures (1.5 × 1.2 m) that were randomly distributed in each site. The clipped vegetation was stored in airtight boxes to avoid respiration losses. The samples were dried in an oven at 60 °C for at least 24 h and weighed. Biomass produced per ha was then calculated. The nitrogen content of leaves was determined using the Kjeldahl technique to determine crude protein (CP) (CP% = N% × 6.25). Near-infrared spectroscopy (NIRS) techniques as described by Corson et al. [25] were used in analyzing the nutritive forage values of organic matter and neutral detergent fiber (NDF) percentage content. The in vitro dry matter digestibility (IVDMD) was estimated using the two-stage technique of Tilley and Terry [26]. The metabolizable energy (ME) was calculated using the equations of Menke and Steingass [27].

### 2.3. Selection of Explanatory Variables for Modelling

In the studied ecosystems, the annual net primary production (ANPP) varied from 3.5 g/m$^2$/yr for overgrazed grassland in the Mata Negra thicket to 58.5 g/m$^2$/yr under moderate grazing in Andean grasslands (Table 1). Overgrazing reduced ANPP by two thirds in most ecosystems. Forage quality also showed a great variation among sites (Table 1). In the lamb and wool production calculation, we estimated from forage production and quality characteristics a mean intake of 2–3% of their live weight in DM daily [28]. We also calculated a mean conversion efficiency of 42.6 g of forage into 1 g of live carcass, and a mean conversion efficiency of 140.2 g of forage into 1 g of wool [23,28].

**Table 1.** Plant forage production and quality on dry matter (DM) basis in the evaluated sites in Santa Cruz Province.

| Variable | Range |
|---|---|
| Annual net primary production (ANPP) | 3.5–58.5 g DM/m$^2$/yr |
| Organic matter yield | 3.2–53.2 g DM/m$^2$/yr |
| Crude protein | 8.8–17.2% |
| In vitro dry matter digestibility | 49.7–61.8% |
| Metabolizable energy | 1.79–2.23 Mcal/kg DM |
| Neutral detergent fiber | 34.8–50.3% |

To build the live weight lamb and wool production models, first we explored 28 potential explanatory variables for the 120 evaluated sites (Table 2). These variables were rasterized at 90 × 90 m resolution in a geographical information system (GIS) using the nearest resampling technique on ArcMap 10.0 software [29]. This resampling allowed us to obtain grids with the same pixel size and compatible formats for the further analyses. Climatic variables (*n* = 21) [30] included temperature, precipitation, and indexes of annual, monthly, or seasonal variations, global potential evapotranspiration, and global aridity index [31]. Topography variables (*n* = 4) included elevation, slope [32], and aspect [33]. Landscape metrics (*n* = 3) included the normalized difference vegetation index (NDVI) [34], net primary productivity (ANPP) [35] and desertification index in six categories (0 = none, 1 = slight degraded, 2 = moderate desertification, 3 = moderate to severe desertification, 4 = severe desertification, 5 = very severe desertification) [18]. A pre-selection of the potential variables to be included in the models was performed based on Pearson's correlation indices. This index allowed us to obtain paired analyses of each dataset, considering the strength of the linear relationship (−1 to +1), and a *p*-value of less than 0.05 with a confidence level of 95%.

**Table 2.** Explanatory variables used in the animal and wool yield analysis.

| Category | Description | Code | Unit | Data Source |
|---|---|---|---|---|
| Climate | mean annual temperature | AMT | °C | WorldClim [1] |
| | mean diurnal range | MDR | °C | WorldClim [1] |
| | isothermality | ISO | % | WorldClim [1] |
| | temperature seasonality | TS | °C | WorldClim [1] |
| | max temperature of warmest month | MAXWM | °C | WorldClim [1] |
| | min temperature of coldest month | MINCM | °C | WorldClim [1] |
| | temperature annual range | TAR | °C | WorldClim [1] |
| | mean temperature of wettest quarter | MTWEQ | °C | WorldClim [1] |
| | mean temperature of driest quarter | MTDQ | °C | WorldClim [1] |
| | mean temperature of warmest quarter | MTWAQ | °C | WorldClim [1] |
| | mean temperature of coldest quarter | MTCQ | °C | WorldClim [1] |
| | mean annual precipitation | AP | mm/years | WorldClim [1] |
| | precipitation of wettest month | PWEM | mm/years | WorldClim [1] |
| | precipitation of driest month | PDM | mm/years | WorldClim [1] |
| | precipitation seasonality | PS | % | WorldClim [1] |
| | precipitation of wettest quarter | PWEQ | mm/years | WorldClim [1] |
| | precipitation of driest quarter | PDQ | mm/years | WorldClim [1] |
| | precipitation of warmest quarter | PWAQ | mm/years | WorldClim [1] |
| | precipitation of coldest quarter | PCQ | mm/years | WorldClim [1] |
| | global potential evapo-transpiration | EVTP | mm/years | CSI [2] |
| | global aridity index | GAI | | CSI [2] |
| Topography | elevation | ELE | m.a.s.l. | DEM [3] |
| | slope | SLO | % | DEM [3] |
| | aspect cosine | ASPC | cosine | DEM [3] |
| | aspect sine | ASPS | sine | DEM [3] |
| Landscape | normalized difference vegetation index | NDVI | dimensionaless | MODIS [4] |
| | net primary productivity | NPP | $\mathrm{gr\ C.m^2/year}$ | MODIS [5] |
| | desertification | DES | dimensionaless | CENPAT [6] |

[1] Hijmans et al. [30], [2] Consortium for Spatial Information (CSI) [31], [3] Farr et al. [32], [4] ORNL DAAC [34], [5] Zhao et al. [35], [6] Del Valle et al. [18].

## 2.4. Modelling and Calibration

We used stepwise multiple regressions to identify which variables among these uncorrelated variables helped to explain lamb (g lamb/m$^2$/yr) and greasy wool yield (g greasy wool/m$^2$/yr) at a landscape level. The selected explanatory variables must present a low

correlation with the other potential selected variables, despite the correlation of the independent variables (lamb and greasy wool yield). The regression analyses were performed through the ordinary least squares (OLS) method, minimizing the sum of square differences between the observed and predicted values. We employed $p < 0.05$ for the significance probability for each regression statistic included in the model, and used 500 steps for the final model selection. The models were evaluated through the adjustment ($r^2$-adj), the standard error (SE) of estimation defined as the average of the difference between predicted versus observed values, and the mean absolute error (AE) defined as the average of the difference between predicted versus the observed absolute values (Statgraphics Centurion software, Statpoint Technologies, The Plains, VA, USA).

To test the model, we performed a calibration procedure using the same database employed for the modelling (observed vs. modelled). The test was carried out by analyzing the mean and absolute errors (differences between observed and modelled values) of liveweight lamb and greasy wool animal expressed as g lamb/m$^2$/yr and g greasy wool/m$^2$/yr. These analyses allowed us to describe the distribution of the errors across the different studied gradients.

With the adjusted models, we produced two final maps for the entirety of Santa Cruz Province (Argentina), where the variables derived from the multiple linear regression models were integrated into the GIS using ArcMap 10.0 software. These maps were adjusted to better represent the livestock activities. For this, we applied a mask to remove areas with: (i) NDVI < 0.05 that included glaciers, water bodies, rocks, and areas without vegetation cover [36], (ii) ELE > 1200 m.a.s.l. where sheep production was not conducted due to extreme climate, (iii) *Nothofagus pumilio* and mixed evergreen forests, and (iv) natural protected networking areas. Finally, we analyzed the two maps considering the main ecological areas to determine differences among the studied categories. For this, we extract the values of each map using a hexagonal binning processes (each hexagon = 250,000 ha), and the values were compared through one-way ANOVAs and a Tukey post hoc test. These outputs allowed us to characterize the maps across the landscape.

## 3. Results

Across Santa Cruz province, animal yield ranged from 0.25 to 0.69 g lamb/m$^2$/yr and 0.10 to 0.19 g greasy wool/m$^2$/yr. Most of the variables were highly correlated with lamb and wool yields using the Pearson's correlation index (Table A1), where NDVI (0.665–0.699, $p < 0.001$) was the most correlated.

The stepwise multiple regression selected three variables for the modelling of lamb yield: isothermality (ratio of average day variation in temperature divided by annual variability in temperature) (ISO,%), normalized vegetation index (NVDI, dimensionless), and desertification index (DES, categories 0 = none, 1 = slight degraded, 2 = moderate desertification, 3 = moderate to severe desertification, 4 = severe desertification, 5 = very severe desertification). The fitted model ($r^2$-adj = 0.960; F = 960.8; SE = 0.084; AE = 0.064) explained 96.0% of variation in lamb yield values, and was expressed as:

$$\text{Lamb yield (g lamb/m}^2\text{/yr)} = 0.00854539 \times \text{ISO} + 0.309721 \times \text{NDVI} - 0.0263287 \times \text{DES} \quad (1)$$

The most important variables for the model of greasy wool production were isothermality (ratio of average day variation in temperature divided by annual variability in temperature) (ISO,%) and normalized vegetation index (NVDI, dimensionless). The fitted model ($r^2$-adj = 0.984; F = 3643.6; SE = 0.017; AE = 0.014) explained 98.4% of variation in greasy wool yield and had the following formula:

$$\text{Wool yield (g greasy wool/m}^2\text{/yr)} = 0.00248603 \times \text{ISO} + 0.0756224 \times \text{NDVI} \quad (2)$$

In Tables A2 and A3, the main descriptive statistics of the explanatory variables and model outputs used for lamb and wool modelling yields analysis are presented. When

univariate correlations were performed, these variables correlated strongly with lamb and wool yields and there was no evidence of collinearity between them ($p < 0.001$).

The map of the adjusted lamb yield model across Santa Cruz showed high values in the ecotone between *Nothofagus antarctica* forest and grasslands in the west, in the south and in river valleys and wetlands where most productive rangelands dominate, and low values in the northeast and central areas of the province (Figure 2). A similar pattern but with less magnitude in values was observed for greasy wool yield (Figure 3).

When the performance of the model was tested across different climate and management related gradients, it can be observed that the error dispersion was quite homogeneous (Table 3). In general, errors increased with lamb and wool yields.

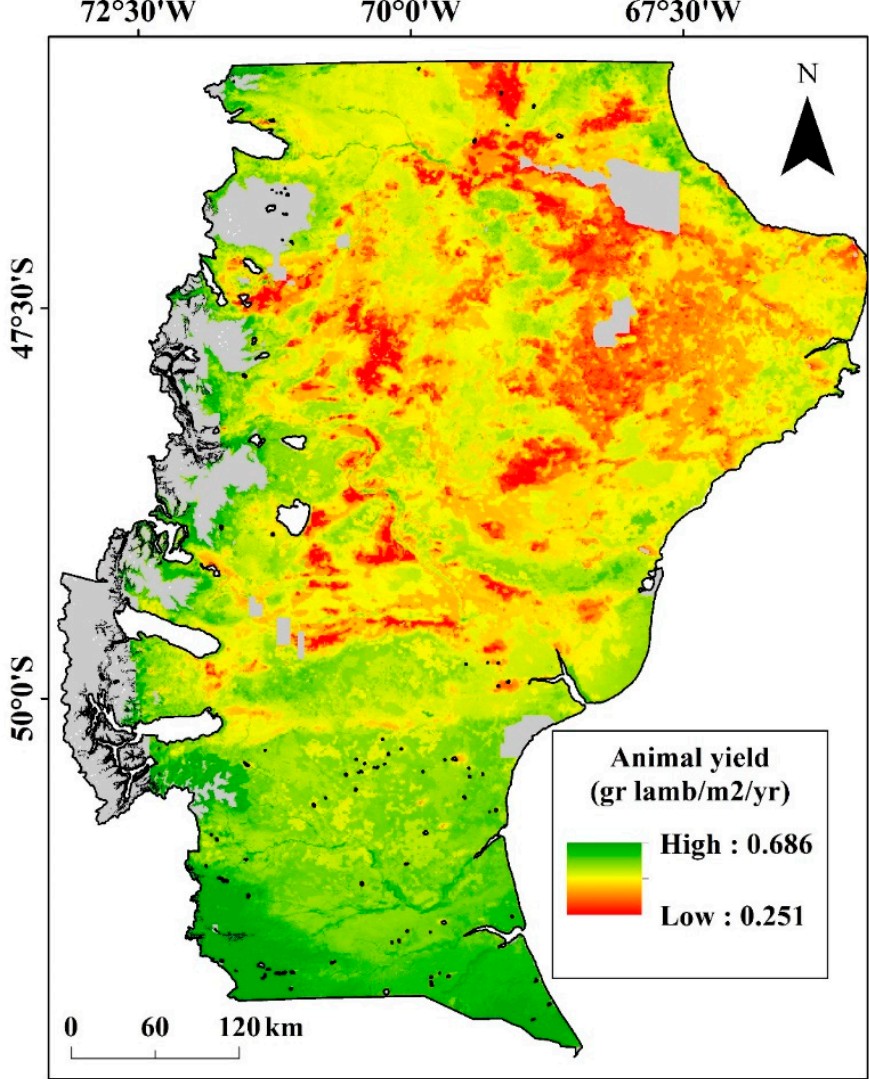

**Figure 2.** Lamb yield map varied from 0.251 (red) to 0.686 (green) gr lamb/m$^2$/yr. Areas without livestock activities (value = 0) are identified in gray (NDVI < 0.05, ELE > 1200 m.a.s.l and protected areas) and in black (*Nothofagus pumilio* and mixed evergreen forest).

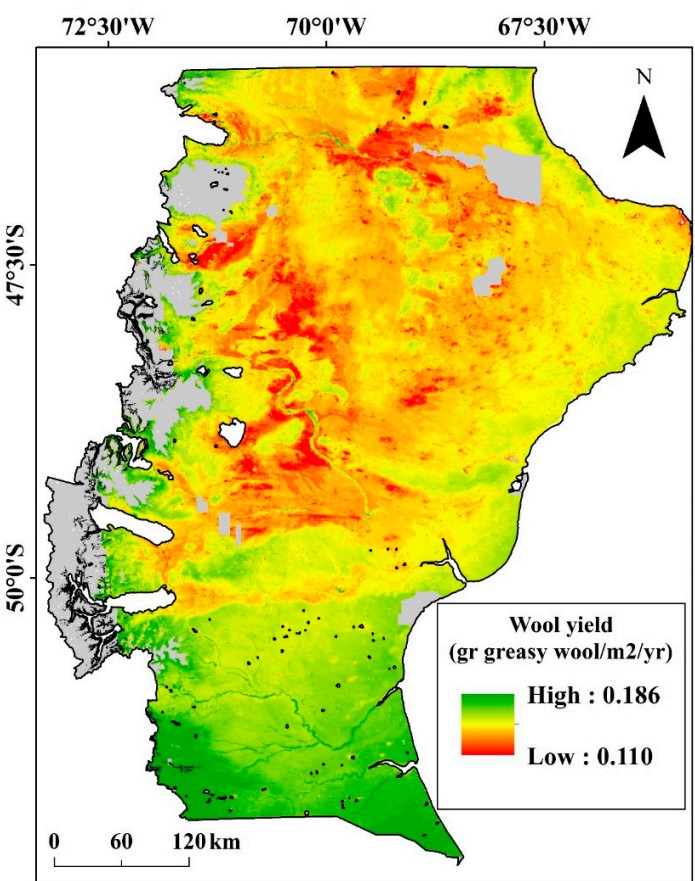

**Figure 3.** Wool yield map varied from 0.110 (red) to 0.186 (green) g greasy wool/m$^2$/yr. Areas without livestock activities (value = 0) are identified in gray (NDVI < 0.05, ELE > 1200 m.a.s.l and protected areas) and in black (*Nothofagus pumilio* and mixed evergreen forest).

**Table 3.** Model performance analyses (data in the field vs. modelled) using a calibration of lamb and greasy wool yields sorted by discrete variable categories covering the Santa Cruz province for the selected variables in the modelling: (i) temperature seasonality, (ii) normalized vegetation index (NVDI), (iii) desertification index. N = number of plots, ME = mean error, AE = absolute error.

| Selected Variables | N | Lamb Yield (gr lamb/m$^2$/yr) | Modelled | ME | AE |
|---|---|---|---|---|---|
| Temperature Seasonality (°C) | | | | | |
| <46 | 46 | 0.40 | 0.37 | 0.03 | 0.07 |
| 46–47 | 41 | 0.38 | 0.38 | −0.01 | 0.06 |
| >47 | 33 | 0.46 | 0.49 | −0.03 | 0.08 |
| NDVI | | | | | |
| <0.18 | 43 | 0.36 | 0.34 | 0.02 | 0.05 |
| 0.18–0.27 | 37 | 0.37 | 0.38 | −0.01 | 0.08 |
| >0.27 | 40 | 0.49 | 0.51 | −0.02 | 0.08 |
| Desertification | | | | | |
| Low | 42 | 0.51 | 0.50 | 0.00 | 0.07 |
| Moderate | 42 | 0.37 | 0.37 | −0.01 | 0.07 |
| High | 36 | 0.34 | 0.33 | 0.01 | 0.06 |
| Total | 120 | 0.41 | 0.41 | 0.00 | 0.07 |

**Table 3.** *Cont.*

| Selected variables | N | Greasy wool yield (gr greasy wool/m$^2$/yr) | Modelled | ME | AE |
|---|---|---|---|---|---|
| Temperature seasonality (°C) | | | | | |
| <46 | 46 | 0.13 | 0.13 | 0.01 | 0.01 |
| 46–47 | 41 | 0.13 | 0.13 | 0.00 | 0.01 |
| >47 | 33 | 0.14 | 0.15 | −0.01 | 0.01 |
| NDVI | | | | | |
| <0.18 | 43 | 0.13 | 0.12 | 0.00 | 0.01 |
| 0.18–0.27 | 37 | 0.13 | 0.13 | 0.00 | 0.02 |
| >0.27 | 40 | 0.15 | 0.15 | 0.00 | 0.02 |
| Total general | 120 | 0.14 | 0.14 | 0.00 | 0.01 |

There were differences in lamb production across vegetation types with mean values that varied from 0.35 g lamb/m$^2$/yr in grasslands in the Central plateau to 0.51 g lamb/m$^2$/yr in the *Nothofagus antarctica* forest (Table 4). Greasy wool yield ranged from 0.13 (Mata Negra thicket and plateaus) to 0.15 g greasy wool/m$^2$/yr (Magellanic grass steppe and *N. antarctica*). Finally, lamb and greasy wool yields decreased with desertification gradient (Table 4) due to erosion processes.

**Table 4.** Simple ANOVA analyses of lamb and greasy wool yields comparing the outputs of the models across the different ecological areas and different levels of desertification in Santa Cruz Province. N = number of hexagons extracted in the SIG for the different categories.

| Ecological Areas | N | Lamb Yield (gr lamb/m$^2$/yr) | Greasy Wool Yield (gr greasy wool/m$^2$/yr) |
|---|---|---|---|
| Central plateau | 77 | 0.35 a | 0.13 a |
| Mata negra thicket | 13 | 0.40 b | 0.13 b |
| Sub-Andean vegetation | 9 | 0.43 b | 0.14 b |
| Magellanic grass steppe | 11 | 0.48 c | 0.15 c |
| *Nothofagus antarctica* forests | 10 | 0.51 c | 0.15 c |
| F(*p*) | 120 | 33.78 (<0.001) | 44.67 (<0.001) |
| **Desertification** | **N** | **Lamb yield (gr lamb/m$^2$/yr)** | **Greasy wool yield (gr greasy wool/m$^2$/yr)** |
| High | 40 | 0.33 a | 0.12 a |
| Moderate | 37 | 0.36 b | 0.13 a |
| Low | 43 | 0.45 c | 0.14 b |
| F(*p*) | 120 | 129.43 (<0.001) | 90.00 (<0.001) |

F = Fisher test, (*p*) probability. Different letters showed differences with Tukey test at *p* < 0.05.

## 4. Discussion

Variability in lamb and wool production between farms can be attributed to differences in grassland condition (forage quantity and quality) that in turn can be related to long term grazing management and climate conditions. In this work, fitted models for lamb and wool yield predictions were able to account for 95% of the variation across the study area with values ranging from 0.25 to 0.69 g lamb/m$^2$/yr and 0.10 to 0.19 g greasy wool/m$^2$/yr. These results confirm the hypothesis that lamb production would be more sensitive than wool production to environmental conditions and forage quantity at the regional scale in Patagonia. In this study, vegetation cover, as represented by the Normalized Difference

Vegetation Index (NDVI), was a strong predictor of lamb and wool yields in the fitted models. Values of NDVI may act as a surrogate for steppe ANPP that is affected by grazing management and climate. This was consistent with Piñeiro et al. [37] who reported that NDVI was correlated with ANPP in grasslands. Similarly, Paruelo et al. [38] reported that the spatial and temporal patterns of the Patagonian steppe ecosystem functioning was described from an empirical relationship between the Landsat TM-derived NDVI and field ANPP. In semi-arid and arid Central Asia, the NDVI-rainfall relation was correlated with livestock density, reinforcing the conclusion that NDVI provides a proxy for spatial and temporal vegetation biomass allowance [39].

Mapping and modelling of ES is one of the topics within the ES research field that has gained much attention in recent years [40]. The methods of mapping livestock provisioning ES in the present study (lamb and wool), may be integrated into decision support systems, making them available to the wider public and decision makers. However, the main weakness of the modelling was the limited availability of data at the landscape level. For example, we used climate and topography as the main explanatory variables, and only three variables directly related to the productivity (NDVI, NPP and desertification). There are some other factors that can directly influence livestock production (e.g., animal comfort, water availability, predators) or indirectly through grasslands productivity at a local scale (e.g., soil types, fertility, soil water content) [1,2,4,18,23]. However, these factors are not available in mapped information to be included in modelling [9,10]. In this context, the present model is the best option to estimate lamb and wool production in Patagonia [14,24].

Production from grassland in good ecological conditions in the Andean grasslands with *N. antarctica* forest had significantly higher animal production values for lamb and wool than production on overgrazed and ecologically degraded sites in the less productive Central Plateau grasslands. Thus, the climate and the vegetation influenced animal provisioning ES. These differences in the productivity of livestock responded to the quality-adjusted yield as it represents an integrated measure of biomass yield and forage quality, particularly the metabolizable energy because it is considered a useful measure for overall ruminant-specific nutritional value as a main limiting factor for ruminant production [41].

However, in Patagonia over the last 70 years, we have witnessed extensive degradation of once productive steppe ecosystems (desertification) [42]. Thus, heavy and unsustainable grazing conditions threaten the future of livestock productivity, therefore threatening the long-term wellbeing of the local economy [15,16]. This has impacted on farming development and sheep stocks have declined by half since the early twentieth century, and more than 500 farms have been abandoned [43,44]. Significant structural changes are required to reach sustainable Patagonian grassland use, together with competitive and diversified products of high quality.

The results of lamb and wool production found in this study based on grassland forage quantity and quality and animal requirements may assist stakeholders and policy makers to adopt sustainable management practices aiming to increase animal production efficiency. Extensive livestock systems common in Patagonia result in low production efficiency where irregular use of rangelands by domestic herbivores is one of the most common problems facing grazing management [45,46]. The extensive system in South Patagonia continues to be carried out mainly under a scheme of minimal intervention of the landscape [47]. Policy decisions may play an important role in arid and semi-arid environments by providing tools and alternatives to improve animal yield. In Patagonia, we can improve lamb output per ewe by selective breeding that improves ewe productivity, increasing lamb survival through better management at birth and by improving nutritional management. A more uniform use of the rangelands at moderate stocking rates together with supplementation strategies, subdivision of paddocks, periodic herding, or the provisioning of new water sources would increase forage and animal productivity and plant species richness, attenuate the degradation of the most preferred sites, and improve the health of the rangeland systems [48–50]. For example, the program developed into the National Sheep Evaluation Service (PROVINO) based on a joint agreement between the

National Institute for Agricultural Technology (INTA) and several breed societies provides fleece testing based on body weight, fleece weight, clean yield, and fiber diameter. Another important development for the Argentine sheep industry has been the enforcement of a "National Sheep Recovery Law". This Law provides state funds for stock recovery, farm infrastructure, feed production, breeding plans, large scale sheep health programs, semen imports, and central progeny testing. This law also finances the National Wool Quality Program called "Prolana", which promotes and rewards proper shearing, skirting, baling, and testing of wool. Furthermore, in dry cold environments like Patagonia, management emphasis should encompass resiliency, risk reduction, avoidance of grassland degradation, and low input for sustainability. For this, it may be necessary to review the technology transfer system from research and extension institutions.

Apart from their primary function of producing lamb and wool, most sheep farming systems in Patagonia provide other benefits to society, such as biodiversity conservation, regulating services (e.g., erosion and climate control), supporting services (e.g., nutrient cycling), and cultural services (e.g., recreation, local identity, tourism) [11–14]. Most ES go unrecognized and undervalued in economic decisions (production and transaction), government policies, and management practices where historically, markets have largely focused on provisioning services (e.g., meat products, wool). Among other outcomes, decision-making aims at strengthening synergies and minimizing trade-offs among ecosystem services and among ES beneficiaries at different spatial scales [51]. In many cases, an increase in one ES (e.g., food production) can negatively affect the provision of another ES (e.g., drinking water quality), which represents a trade-off among multiple ES, while an increase in one ES (e.g., honey production) can positively affects the provision of another ES (e.g., fruit production), which can usually be perceived as synergetic. Thus, although lamb and wool provisioning services are likely the best recognized ES in Patagonian grasslands since they contribute directly to human material wellbeing, other supporting services that contribute indirectly to human wellbeing by maintaining the processes and functions necessary for provisioning, regulating, and cultural services should also be incorporated into decision-making from stakeholders [52].

## 5. Conclusions

This study has provided data on lamb and wool production in Southern Patagonia at a regional scale including the principal ecosystem types found in the Patagonian rangeland. We found that lamb production was more sensitive than wool production to grassland condition (forage quantity and quality) between farms. This can be attributed to differences in long term grazing management and climate conditions. The successful management of livestock is an important challenge for the commercial and policy communities to satisfy society's needs for food and wool products under sustainable grassland management. The results of lamb and wool production found in the present work assist to characterize the provisioning ES ecosystem of livestock products in Southern Patagonia by providing a baseline against which management actions can be planned and progress monitored.

**Author Contributions:** Conceptualization, P.L.P.; formal analysis, E.R.; investigation, P.L.P. and G.M.P.; methodology, P.L.P. and Y.M.R. All authors have read and agreed to the published version of the manuscript.

**Funding:** This research received no external funding.

**Institutional Review Board Statement:** Not applicable.

**Informed Consent Statement:** Not applicable.

**Conflicts of Interest:** The authors declare no conflict of interest.

# Appendix A

**Table A1.** Pearson's correlation index among exploratory variables used in lamb (LY) and wool (WY) yield analysis. (See Table 1 for variable definitions). Indexes printed in bold are significant ($p < 0.05$).

| | LY | WY | AMT | MDR | ISO | TS | MAXWM | MINCM | TAR | MTWEQ | MTDQ | MTWAQ | MTCQ | AP | PWEM | PDM | PS | PWEQ | PDQ | PWAQ | PCQ | GAI | EVTP | ELE | SLO | ASPS | ASPC | PPN | NDVI |
|---|---|---|---|---|---|---|---|---|---|---|---|---|---|---|---|---|---|---|---|---|---|---|---|---|---|---|---|---|---|
| WY | 0.98 | | | | | | | | | | | | | | | | | | | | | | | | | | | | |
| AMT | −0.52 | −0.54 | | | | | | | | | | | | | | | | | | | | | | | | | | | |
| MDR | −0.55 | −0.56 | 0.42 | | | | | | | | | | | | | | | | | | | | | | | | | | |
| ISO | 0.18 | 0.19 | −0.53 | −0.12 | | | | | | | | | | | | | | | | | | | | | | | | | |
| TS | −0.60 | −0.61 | 0.68 | 0.82 | −0.63 | | | | | | | | | | | | | | | | | | | | | | | | |
| MAXWM | −0.56 | −0.59 | 0.97 | 0.61 | −0.56 | 0.83 | | | | | | | | | | | | | | | | | | | | | | | |
| MINCM | −0.33 | −0.36 | 0.90 | 0.06 | −0.34 | 0.32 | 0.78 | | | | | | | | | | | | | | | | | | | | | | |
| TAR | −0.55 | −0.56 | 0.59 | 0.90 | −0.54 | 0.98 | 0.77 | 0.21 | | | | | | | | | | | | | | | | | | | | | |
| MTWEQ | −0.17 | −0.17 | 0.05 | 0.32 | 0.36 | 0.10 | 0.07 | −0.01 | 0.12 | | | | | | | | | | | | | | | | | | | | |
| MTDQ | −0.32 | −0.35 | 0.81 | 0.28 | −0.80 | 0.70 | 0.82 | 0.68 | 0.60 | −0.19 | | | | | | | | | | | | | | | | | | | |
| MTWAQ | −0.55 | −0.57 | 0.98 | 0.53 | −0.60 | 0.80 | 0.99 | 0.82 | 0.72 | 0.06 | 0.85 | | | | | | | | | | | | | | | | | | |
| MTCQ | −0.40 | −0.43 | 0.96 | 0.20 | −0.42 | 0.46 | 0.87 | 0.98 | 0.37 | 0.02 | 0.73 | 0.90 | | | | | | | | | | | | | | | | | |
| AP | 0.57 | 0.54 | −0.38 | −0.61 | 0.02 | −0.53 | −0.45 | −0.16 | −0.54 | −0.22 | −0.15 | −0.42 | −0.25 | | | | | | | | | | | | | | | | |
| PWEM | 0.51 | 0.48 | −0.28 | −0.57 | −0.07 | −0.43 | −0.35 | −0.10 | −0.46 | −0.18 | −0.03 | −0.31 | −0.17 | 0.97 | | | | | | | | | | | | | | | |
| PDM | 0.58 | 0.55 | −0.39 | −0.58 | 0.12 | −0.58 | −0.46 | −0.17 | −0.55 | −0.20 | −0.23 | −0.45 | −0.26 | 0.96 | 0.90 | | | | | | | | | | | | | | |
| PS | −0.24 | −0.26 | 0.45 | 0.14 | −0.54 | 0.48 | 0.45 | 0.34 | 0.36 | −0.08 | 0.63 | 0.49 | 0.37 | −0.18 | 0.02 | −0.39 | | | | | | | | | | | | | |
| PWEQ | 0.53 | 0.50 | −0.29 | −0.63 | −0.07 | −0.48 | −0.38 | −0.08 | −0.51 | −0.28 | −0.04 | −0.34 | −0.17 | 0.98 | 0.98 | 0.91 | −0.01 | | | | | | | | | | | | |
| PDQ | 0.57 | 0.55 | −0.38 | −0.57 | 0.07 | −0.55 | −0.45 | −0.17 | −0.53 | −0.19 | −0.20 | −0.43 | −0.25 | 0.97 | 0.91 | 0.99 | −0.36 | 0.92 | | | | | | | | | | | |
| PWAQ | 0.54 | 0.52 | −0.52 | −0.54 | 0.40 | −0.68 | −0.59 | −0.28 | −0.64 | 0.01 | −0.48 | −0.58 | −0.37 | 0.89 | 0.82 | 0.93 | −0.48 | 0.82 | 0.92 | | | | | | | | | | |
| PCQ | 0.50 | 0.47 | −0.23 | −0.63 | −0.18 | −0.42 | −0.32 | −0.04 | −0.46 | −0.40 | 0.06 | −0.27 | −0.12 | 0.95 | 0.95 | 0.87 | 0.04 | 0.98 | 0.88 | 0.73 | | | | | | | | | |
| GAI | 0.62 | 0.60 | −0.50 | −0.66 | 0.11 | −0.62 | −0.57 | −0.28 | −0.61 | −0.23 | −0.26 | −0.54 | −0.37 | 0.99 | 0.94 | 0.96 | −0.25 | 0.96 | 0.97 | 0.92 | 0.92 | | | | | | | | |
| EVTP | −0.60 | −0.63 | 0.94 | 0.68 | −0.55 | 0.87 | 0.99 | 0.71 | 0.83 | 0.10 | 0.80 | 0.98 | 0.82 | −0.50 | −0.40 | −0.52 | 0.47 | −0.43 | −0.50 | −0.63 | −0.37 | −0.62 | | | | | | | |
| ELE | 0.11 | 0.13 | −0.22 | 0.18 | −0.46 | 0.34 | −0.06 | −0.44 | 0.36 | −0.14 | 0.25 | −0.08 | −0.37 | 0.14 | 0.21 | 0.07 | 0.27 | 0.15 | 0.10 | −0.10 | 0.19 | 0.15 | −0.01 | | | | | | |
| SLO | 0.06 | 0.06 | −0.24 | −0.05 | −0.01 | −0.05 | −0.20 | −0.26 | −0.04 | −0.05 | −0.10 | −0.20 | −0.26 | 0.28 | 0.28 | 0.24 | −0.01 | 0.27 | 0.25 | 0.24 | 0.27 | 0.28 | −0.18 | 0.19 | | | | | |
| ASPS | 0.10 | 0.09 | −0.14 | −0.11 | 0.15 | −0.20 | −0.15 | −0.07 | −0.16 | 0.01 | −0.15 | −0.16 | −0.10 | 0.14 | 0.11 | 0.18 | −0.16 | 0.12 | 0.17 | 0.17 | 0.10 | 0.17 | −0.15 | −0.01 | 0.08 | | | | |
| ASPC | 0.04 | 0.02 | 0.02 | 0.14 | 0.01 | 0.09 | 0.05 | −0.04 | 0.11 | 0.14 | 0.02 | 0.04 | −0.01 | −0.04 | −0.03 | −0.01 | −0.09 | −0.07 | −0.01 | 0.01 | −0.10 | −0.04 | 0.04 | −0.03 | 0.05 | −0.06 | | | |
| PPN | 0.55 | 0.51 | −0.36 | −0.51 | 0.29 | −0.61 | −0.44 | −0.12 | −0.56 | −0.15 | −0.29 | −0.43 | −0.21 | 0.68 | 0.60 | 0.72 | −0.35 | 0.63 | 0.71 | 0.72 | 0.58 | 0.70 | −0.49 | −0.07 | 0.10 | 0.28 | −0.04 | | |
| NDVI | 0.69 | 0.66 | −0.66 | −0.53 | 0.40 | −0.72 | −0.70 | −0.46 | −0.63 | −0.13 | −0.49 | −0.70 | −0.54 | 0.71 | 0.62 | 0.75 | −0.44 | 0.63 | 0.74 | 0.77 | 0.58 | 0.78 | −0.73 | 0.07 | 0.17 | 0.23 | −0.02 | 0.84 | |
| DES | −0.65 | −0.64 | 0.52 | 0.45 | −0.30 | 0.55 | 0.56 | 0.37 | 0.51 | −0.03 | 0.42 | 0.55 | 0.43 | −0.57 | −0.56 | −0.53 | 0.11 | −0.55 | −0.53 | −0.59 | −0.49 | −0.61 | 0.57 | −0.05 | −0.14 | −0.14 | −0.09 | −0.49 | −0.66 |

**Table A2.** Descriptive statistics of the explanatory variables in the analyses.

| Category | Variable | Mean | Median | Standard Deviation | Max | Min | Standard Error |
|---|---|---|---|---|---|---|---|
| Climate | AMT | 8.30 | 8.20 | 2.01 | 13.20 | 3.80 | 0.18 |
| | MDR | 10.21 | 10.40 | 0.71 | 11.20 | 8.50 | 0.07 |
| | ISO | 46.43 | 46.00 | 1.81 | 51.00 | 44.00 | 0.17 |
| | TS | 4.38 | 4.34 | 0.48 | 5.20 | 3.46 | 0.04 |
| | MAXWM | 19.98 | 20.45 | 2.88 | 26.60 | 14.10 | 0.26 |
| | MINCM | −1.84 | −2.10 | 1.87 | 2.50 | −5.50 | 0.17 |
| | TAR | 21.82 | 21.60 | 1.83 | 24.80 | 18.10 | 0.17 |
| | MTWEQ | 6.92 | 5.65 | 3.38 | 14.20 | 0.20 | 0.31 |
| | MTDQ | 10.57 | 11.40 | 4.06 | 16.90 | 4.40 | 0.37 |
| | MTWAQ | 13.66 | 13.80 | 2.55 | 19.80 | 8.10 | 0.23 |
| | MTCQ | 2.50 | 2.15 | 1.72 | 6.70 | −1.10 | 0.16 |
| | AP | 245.12 | 204.00 | 121.73 | 870.00 | 146.00 | 11.11 |
| | PWEM | 30.54 | 26.00 | 13.70 | 98.00 | 17.00 | 1.25 |
| | PDM | 12.98 | 11.00 | 7.75 | 51.00 | 4.00 | 0.71 |
| | PS | 26.03 | 25.00 | 7.51 | 65.00 | 13.00 | 0.69 |
| | PWEQ | 81.84 | 71.00 | 40.43 | 287.00 | 44.00 | 3.69 |
| | PDQ | 44.24 | 36.00 | 25.11 | 170.00 | 18.00 | 2.29 |
| | PWAQ | 53.53 | 43.50 | 28.66 | 186.00 | 20.00 | 2.62 |
| | PCQ | 68.35 | 57.50 | 35.20 | 242.00 | 39.00 | 3.21 |
| | EVTP | 815.80 | 827.50 | 97.59 | 1021.00 | 619.00 | 8.91 |
| | GAI | 0.31 | 0.24 | 0.19 | 1.26 | 0.17 | 0.02 |
| Topography | ELE | 325.08 | 264.50 | 244.27 | 933.00 | 12.00 | 22.30 |
| | SLO | 4.29 | 3.51 | 3.12 | 15.14 | 0.41 | 0.28 |
| | ASPC | 0.13 | 0.26 | 0.72 | 1.00 | −1.00 | 0.07 |
| | ASPS | 0.04 | 0.10 | 0.68 | 1.00 | −1.00 | 0.06 |
| Landscape and land-use | NDVI | 0.27 | 0.21 | 0.17 | 0.87 | 0.02 | 0.02 |
| | PPN | 172.90 | 152.45 | 111.17 | 758.10 | 56.50 | 10.15 |
| | DES | 2.78 | 3.00 | 1.32 | 5.00 | 0.00 | 0.12 |

**Table A3.** Output of the descriptive statistics of the variables using in the modelling of lamb (LY) and wool (WY) yield analysis.

| Lamb Yield (gr lamb/m$^2$/yr) | | | | | |
|---|---|---|---|---|---|
| **Coefficients** | | | | | |
| Parameter | Estimate | Standard Error | T Statistic | *p*-Value | |
| ISO | 0.00854539 | 0.008 | 11.084 | <0.001 | |
| NDVI | 0.309721 | 0.062 | 5.030 | <0.001 | |
| DES | −0.0263287 | 0.008 | −3.392 | =0.001 | |
| **Analysis of variance** | | | | | |
| Source | Sum of Squares | Df | Mean Square | F-Ratio | *p*-Value |
| Model | 20.794 | 3 | 6.931 | 960.810 | <0.001 |
| Residual | 0.844 | 117 | 0.007 | | |
| Total | 21.639 | 120 | | | |
| Wool yield (gr greasy wool/m$^2$/yr) | | | | | |
| **Coefficients** | | | | | |
| Parameter | Estimate | Standard Error | T Statistic | *p*-Value | |
| ISO | 0.00248603 | 0.001 | 38.228 | <0.001 | |
| NDVI | 0.0756224 | 0.009 | 8.021 | <0.001 | |
| **Analysis of variance** | | | | | |
| Source | Sum of Squares | Df | Mean Square | F-Ratio | *p*-Value |
| Model | 2.241 | 2 | 1.121 | 3643.580 | <0.001 |
| Residual | 0.036 | 118 | 0.000 | | |
| Total | 2.277 | 120 | | | |

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
