# Peer review of "Lamb and Wool Provisioning Ecosystem Services in Southern Patagonia"

_sustainability, doi:10.3390/su13158544_

Round 1

Reviewer 1 Report

Dear Authors,

The article "Lamb and wool provisioning ecosystem services in Southern Patagonia", cover the topic of lamb and wool production as provisioning ecosystem services of livestock products in Santa Cruz province in Southern Patagonia.  The objective of this study was to model lamb and wool production as provisioning ecosystem services at a regional scale using climatic, topographic and vegetation variables from sheep farms across Santa Cruz province. It covers a topic relevant to the Journal of Sustainability because the proper management of sheep farming is an important challenge to satisfy society needs of food and wool products under sustainable grassland management.

Strengths:

  • The topic of the paper is important in the context of grassland and sheep farming Patagonian
  • The paper is well structured.
  • The paper is easy to read.

Weakness:

  1. In part 1. entitled Introduction:
  • The Authors refer to several previously conducted own studies and this section could be improved. Before stating the contribution, Authors should add a paragraph reviewing their most relevant research work that have been done so far and highlight the research gap.
  • The Authors hypothesize that, due to overgrazing at the regional scale in Patagonia, animal production would be lower where environmental conditions are harsh (low soil moisture conditions) and grasslands are degraded. This hypothesis is too general and obvious, it should be more specified. Moreover, Authors never mention above hypothesis again (even in the conclusions).
  1. In part 2. Material and Methods:
  • The time range of the research was not provided.
  • In part 2.3 Selection of explanatory variables for modelling the Authors gave the information “A preselection of the potential variables to be included in the models was performed based on Pearson’s correlation indices. This index allowed to obtain paired analyzes of each data, considering the strength of the linear relationship (-1 to +1), and a P-value less than 0.05 with a confidence level of 95%.” The article lacks the explanation about the specifics of a preselection of the potential explanatory This has a significant matter for further research results, because based on “Annex 1. Pearson’s correlation index among exploratory variables used in lamb (LY) and wool (WY) yields analysis. Index printed in bold are significant (p<0.05)”, the relation between variable LY (by mistake in Annex 1 AY was written – first column should be corrected for LY) and potential exploratory variable ISO is insignificantly correlated with explained variable LY (Pearson correlation index is r=0.18 and it isn’t printed in bold). However, later in part 2.4. Modelling and calibration Authors wrote that  the stepwise multiple regression selected three variables for the modelling of lamb  yield, including: isothermality (ratio of average day variation in temperature divided by annual variability in temperature) (ISO, %).
  1. The main conclusions of the research, in section 5, should be presented in an extended form, so that the significance of the research findings is better exposed.

Author Response

General comments

The comments by both referees have been noted and largely acted upon. The manuscript has been carefully revised. It has been improved by providing clarity, data analysis and interpretation.

Reviewers' comments:
Reviewer #1

  1. In part 1. entitled Introduction:
  • The Authors refer to several previously conducted own studies and this section could be improved. Before stating the contribution, Authors should add a paragraph reviewing their most relevant research work that have been done so far and highlight the research gap.

We have added new paragraphs in the Introduction with 2 new references. Also, the gap of knowledge has been indicated.

  • The Authors hypothesize that, due to overgrazing at the regional scale in Patagonia, animal production would be lower where environmental conditions are harsh (low soil moisture conditions) and grasslands are degraded. This hypothesis is too general and obvious, it should be more specified. Moreover, Authors never mention above hypothesis again (even in the conclusions).

The hypothesis has been improved being more specific. This has been now mentioned in the Discussion section.

  1. In part 2. Material and Methods:
  • The time range of the research was not provided.

Time range has been included

  • In part 2.3 Selection of explanatory variables for modelling the Authors gave the information “A preselection of the potential variables to be included in the models was performed based on Pearson’s correlation indices. This index allowed to obtain paired analyzes of each data, considering the strength of the linear relationship (-1 to +1), and a P-value less than 0.05 with a confidence level of 95%.” The article lacks the explanation about the specifics of a preselection of the potential explanatory

ANSWER: Variables of Table 2 are the data available for the study region. We extracted the data pixel by pixel for the the sample sites. Because there are a lot of variables, first we analyzed the correlation indexes among them (lines 156-159 of the material and methods). This has been clarified in the text.

  • This has a significant matter for further research results, because based on “Annex 1. Pearson’s correlation index among exploratory variables used in lamb (LY) and wool (WY) yields analysis. Index printed in bold are significant (p<0.05)”, the relation between variable LY (by mistake in Annex 1 AY was written – first column should be corrected for LY) and potential exploratory variable ISO is insignificantly correlated with explained variable LY (Pearson correlation index is r=0.18 and it isn’t printed in bold).

ANSWER: We corrected the LY in the table. It is true, that you pointed for the ISO in LY, however it was significant for the WY, and for this, we included in the further analyses. The selection of the variables was conducted based on the correlation among the variables, and not for the individual correlation with the explanatory variables. We understand that ISO alone marginally explained the LY, however, inside the model was one of the best choices for the model. We add some explanation in the methods.

  • However, later in part 4. Modelling and calibration Authors wrote that the stepwise multiple regression selected three variables for the modelling of lamb yield, including: isothermality (ratio of average day variation in temperature divided by annual variability in temperature) (ISO, %).

ANSWER: That’s true, these variables presented low correlation among them, and presented the higher behavior when these were included into the model. You can see that all the variables presented a significant coefficient inside the regression modelling. The Pearson´s coefficient test the performance of each variable alone, but this behavior can change when the variable was combined with others. This has been clarified in the text.

  1. The main conclusions of the research, in section 5, should be presented in an extended form, so that the significance of the research findings is better exposed.

The conclusions have been extended based in the research findings

Reviewer 2 Report

Dear authors,

This research paper describes the actual and interesting topic – Lamb and wool provisioning ecosystem services in Southern Patagonia. Thus, authors notice that the main objective of this study was to model lamb and wool production as provisioning ES at a regional scale using climatic, topographic and vegetation variables from sheep farms across Santa Cruz province. Authors argue, that the results of lamb and wool production found in the present work assist to characterize the provisioning ES ecosystem of livestock products in Southern Patagonia. As well, authors notice that the successful management of livestock becomes an important challenge to the commercial and policy communities to satisfy society needs of food and wool products under sustainable grassland management.

And I would like to share with authors the remarks too: it seems important to notice that it would be needed to take more attention on the discussion and conclusions of the study. Thus, when developing sections of "Discussion" and "Conclusions" it would be needed to include to the debate more newest theoretical implications, thus accessing deeper discussion and concluding insights.

Author Response

And I would like to share with authors the remarks too: it seems important to notice that it would be needed to take more attention on the discussion and conclusions of the study. Thus, when developing sections of "Discussion" and "Conclusions" it would be needed to include to the debate more newest theoretical implications, thus accessing deeper discussion and concluding insights.

We incorporated new paragraph related to new importance of the methods to map livestock provisioning ES by integrating these into decision support systems. For this we have incorporated a new reference (2021). Also, we have added new paragraphs in the Introduction with another 2 new references and the gap of knowledge has been indicated. The hypothesis has been improved being more specific following the advice of reviewer #1, and then included in the Discussion section. The conclusions have been extended based in the research findings.

Round 2

Reviewer 1 Report

Dear Authors,

thank you for your answer. In my opinion, the article has been improved and may be published in the present form.